# Quantitative glycoproteomics of human milk and association with atopic disease

Matilda Holm [1¤a¤b], Mayank Saraswat[1¤c], Sakari Joenväärä[1], Antti Seppo[2], R. John Looney[3], Tiialotta Tohmola[1], Jutta Renkonen[1], Risto Renkonen[1‡], Kirsi M. Järvinen[2‡]*

1 Transplantation Laboratory, Haartman Institute, University of Helsinki and Helsinki University Hospital, Helsinki, Finland, 2 Department of Pediatrics, Division of Allergy, Immunology, and Rheumatology, Center for Food Allergy, University of Rochester School of Medicine and Dentistry, Golisano Children's Hospital, Rochester, New York, United States of America, 3 Department of Medicine, Division of Allergy, Immunology, and Rheumatology, University of Rochester School of Medicine and Dentistry, Rochester, New York, United States of America

¤a Current address: Division of Cellular and Clinical Proteomics, Department of Protein Science, KTH Royal Institute of Technology, Stockholm, Sweden
¤b Current address: Department of Biosciences and Nutrition, Karolinska Institute, Science for Life Laboratory, Solna, Sweden
¤c Current address: Department of Laboratory Medicine and Pathology, Mayo Clinic, Rochester, Minnesota, United States of America
‡ These authors are joint senior authors on this work.
* kirsi_jarvinen-seppo@URMC.Rochester.edu

**Data Availability Statement:** The data underlying this article are available at the ProteomeXchange Consortium (http://proteomecentral.proteomexchange.org) via the PRIDE partner

## Abstract

The prevalence of allergic diseases and asthma is increasing rapidly worldwide, with environmental and lifestyle behaviors implicated as a reason. Epidemiological studies have shown that children who grow up on farms are at lower risk of developing childhood atopic disease, indicating the presence of a protective "farm effect". The Old Order Mennonite (OOM) community in Upstate New York have traditional, agrarian lifestyles, a low rate of atopic disease, and long periods of exclusive breastfeeding. Human milk proteins are heavily glycosylated, although there is a paucity of studies investigating the milk glycoproteome. In this study, we have used quantitative glycoproteomics to compare the N-glycoprotein profiles of 54 milk samples from Rochester urban/suburban and OOM mothers, two populations with different lifestyles, exposures, and risk of atopic disease. We also compared N-glycoprotein profiles according to the presence or absence of atopic disease in the mothers and, separately, the children. We identified 79 N-glycopeptides from 15 different proteins and found that proteins including immunoglobulin A1, polymeric immunoglobulin receptor, and lactotransferrin displayed significant glycan heterogeneity. We found that the abundances of 38 glycopeptides differed significantly between Rochester and OOM mothers and also identified four glycopeptides with significantly different abundances between all comparisons. These four glycopeptides may be associated with the development of atopic disease. The findings of this study suggest that the differential glycosylation of milk proteins could be linked to atopic disease.

repository and can be accessed with the dataset identifier PXD026644.

**Funding:** This work was supported by the National Center for Advancing Translational Sciences [Grant Number R21 TR002516 to KJS and RJL]; Founders' Distinguished Professorship in Pediatric Allergy [KJ]; and Pilot Awards from University of Rochester Clinical and Translational Science Institute and Environmental Health Sciences Center Pilot Award [to RJL]. The content is solely the responsibility of the authors and does not necessarily represent the official views of the National Institutes of Health.

**Competing interests:** The authors have declared that no competing interests exist.

# Introduction

Allergic diseases and asthma are becoming more common worldwide. While an increase in environmental risk factors has contributed to the rise in prevalence, there are also associations between early life circumstances and the development of these diseases [1]. The prevalence of allergic disease and asthma is higher in affluent, developed countries and has increased significantly during the past five decades. The rapid changes in prevalence suggest environmental and lifestyle behaviors as a more likely origin than genetics [2]. Childhood atopic disease includes atopic dermatitis, food allergy, asthma, and allergic rhinitis. The role of breastfeeding in the prevention of childhood atopic disease is controversial, with conflicting data reported. The findings of some studies have shown that breastfeeding decreases the risk of developing atopic disease, while other studies indicate that it either increases the risk or does not affect the development of atopic disease. However, evidence suggests that exclusive breastfeeding for around 4 months may confer a protective effect against the development of childhood atopic disease, especially atopic dermatitis and wheeze [3–5].

Epidemiological studies have shown that children who grow up on farms are at a significantly lower risk of developing childhood atopic disease than children from the same area who do not grow up on farms. This protective "farm effect" is due to factors including early-life contact with livestock and animal feed, as well as the consumption of unprocessed cow's milk. The timing of exposure is crucial for this protective effect, with the strongest effects observed for exposures that occurred in or before the first year of life and those that continued up to the fifth year of life [6, 7]. The Old Order Mennonites (OOM) in Upstate New York predominantly live on farms, have large families and home births, and avoid the use of antibiotics. Having more siblings has been associated with a lower prevalence of hay fever, and vaginal delivery at home has been shown to confer the greatest protection against atopic disease. This OOM community has been found to have a very low rate of allergic disease as compared to a sample of the US population of similar age, sex, and race [2]. Further, the overall prevalence of food allergies was significantly lower in the New York OOM community as compared to the general US population. Rates of breastfeeding at 6 and 12 months and rates of exclusive breastfeeding and 3 and 6 months were also significantly higher in the OOM community compared to in New York State [8].

The proteins in human milk are heavily glycosylated, with up to 70% of human milk proteins estimated to be glycosylated [9]. While the human milk proteome, especially the major components of the protein fraction in milk, has been quite extensively studied in comparison [10–12], studies that have investigated the human milk glycoproteome are fewer and have to a large degree focused on characterizing the N-glycoproteins in milk. One study used LC-MS to identify 32 glycoproteins in milk samples from one donor [13], while another study used multiple reaction monitoring to quantify seven milk proteins and their site-specific N-glycosylation from three milk samples [14]. A study by Cao et al. used quantitative glycoproteomics to compare the N-glycoproteins in human colostrum and mature milk and discovered 68 N-glycosylation sites on 38 proteins that were differentially expressed between the groups [15]. The same authors have also compared the N-glycosylation of human milk fat globule membrane proteins between colostrum and mature milk. This study identified 220 N-glycoproteins with 304 N-glycosylation sites that were differentially expressed between the groups [16]. Previous studies investigating the differences in protein or glycoprotein profiles depending on the presence or absence of atopic disease are very few. One previous study compared the milk proteomes between samples from allergic and non-allergic mothers and discovered 19 proteins whose levels differed significantly between the groups, with the authors proposing that these proteins may be linked to allergy and asthma [17].

In this study, we have used quantitative mass spectrometry-based glycoproteomics to analyze the N-glycoprotein profiles of a total of 54 milk samples from Rochester and OOM mothers, two populations with different lifestyles and exposures. The prevalence of allergic disease is higher in Rochester mothers, and Rochester infants are at higher risk of childhood atopic disease as compared to the OOM of the Finger Lakes region in New York. We compared the N-glycopeptide profiles between these two communities as well as between samples according to the presence or absence of atopic disease in the mothers and, separately, the children (regardless of community). Here, we show that the glycosylation of milk proteins varies depending on lifestyle as well as the presence or absence of atopic disease in both children and, separately, mothers. To the best of our knowledge, this is the first study to compare the milk glycoproteome between two communities with different lifestyles and a high or low risk of atopic disease.

## Results

### Study population and design

Milk samples from a total of 54 mothers were analyzed in this study (see the "Materials and methods" section for details). The samples were divided into groups depending on community (Rochester or OOM) and the N-glycopeptide profiles were compared between these groups. The N-glycopeptide profiles were also compared between samples from mothers whose children developed atopic disease in the first 3 years of life (by parent report) and those whose children did not and, separately, between samples from mothers with or without atopic disease, regardless of which community the mothers belonged to. In our study, the prevalence of childhood atopic disease was twice as common in Rochester children (n = 6) than OOM children (n = 3). The prevalence of atopic disease was also higher in Rochester mothers (n = 16) than OOM mothers (n = 9).

### Identified glycoproteins

We identified a total of 79 N-glycopeptides from 16 different proteins. The 15 proteins identified are listed along with their Uniprot ID and the number of identified glycopeptides for each protein in Table 1. For each of the 79 N-glycopeptides, we also identified the N-glycosylation site, peptide sequence, glycan composition, and proposed glycan structure (given in S1 Table). Different glycoforms were identified for multiple proteins, such as the ones presented below.

### Glycan compositions and proposed structures

The proposed structures for the 79 N-glycan compositions identified matched database entries in the GlyTouCan repository (https://glytoucan.org), enabling the provision of the proposed glycan structures for these glycopeptides. The majority of the glycan compositions identified were classified as complex-type N-glycans, although nine high-mannose N-glycans were also identified. Additionally, two glycan compositions were classified as hybrid-type N-glycans, namely H6N3 and H4N3F1. Further, three glycan compositions were classified as monoantennary N-glycans, namely H4N3F2, H4N3F1, and S1H4N3F1. The glycan composition H4N3F1 that was classified as a hybrid-type N-glycan was found at site 156 on lactotransferrin, while the glycan composition H4N3F1 that was classified as a monoantennary N-glycan was found at site 497 on lactotransferrin.

Alpha-S1-casein (CASA1) was found to have nine glycan compositions, all at site 69. In other words, nine different N-glycopeptides were identified that belonged to alpha-S1-casein. Eight of nine glycan compositions were fucosylated, while four were sialylated. One glycan

**Table 1. Summary of results.**

| Protein name | Protein UniProt ID | Number of identified glycopeptides |
|---|---|---|
| Alpha-1-antitrypsin | A1AT_HUMAN | 1 |
| Butyrophilin subfamily 1 member A1 | BT1A1_HUMAN | 1 |
| Alpha-S1-casein | CASA1_HUMAN | 9 |
| Clusterin | CLUS_HUMAN | 2 |
| Chordin-like protein 2 | CRDL2_HUMAN | 1 |
| Fibrinogen gamma chain | FIBG_HUMAN | 2 |
| Hemopexin | HEMO_HUMAN | 1 |
| Haptoglobin | HPT_HUMAN | 1 |
| Immunoglobulin heavy constant alpha 1 | IGHA1_HUMAN | 8 |
| Immunoglobulin heavy constant alpha 2 | IGHA2_HUMAN | 1 |
| Immunoglobulin J chain | IGJ_HUMAN | 1 |
| Alpha-lactalbumin | LALBA_HUMAN | 1 |
| Lactadherin | MFGM_HUMAN | 2 |
| Polymeric immunoglobulin receptor | PIGR_HUMAN | 24 |
| Lactotransferrin | TRFL_HUMAN | 24 |

composition identified, H6N3, was classified a hybrid-type N-glycan, while the others were classified as complex-type N-glycans. For immunoglobulin A1 (IGHA1, IgA1) we identified five different glycan compositions at site 340 and three at site 144. Of these eight glycans, four were classified as complex-type N-glycans, while the remaining four were classified as high-mannose N-glycans. IgA1 with site 340 indicated is shown in Fig 1A, and the five glycan compositions identified at site 340 are shown in Fig 1B. A total of 24 different glycan compositions at four different N-glycosylation sites were identified for polymeric immunoglobulin receptor (pIgR), the majority of which were complex-type N-glycans. The N-glycosylation site with the

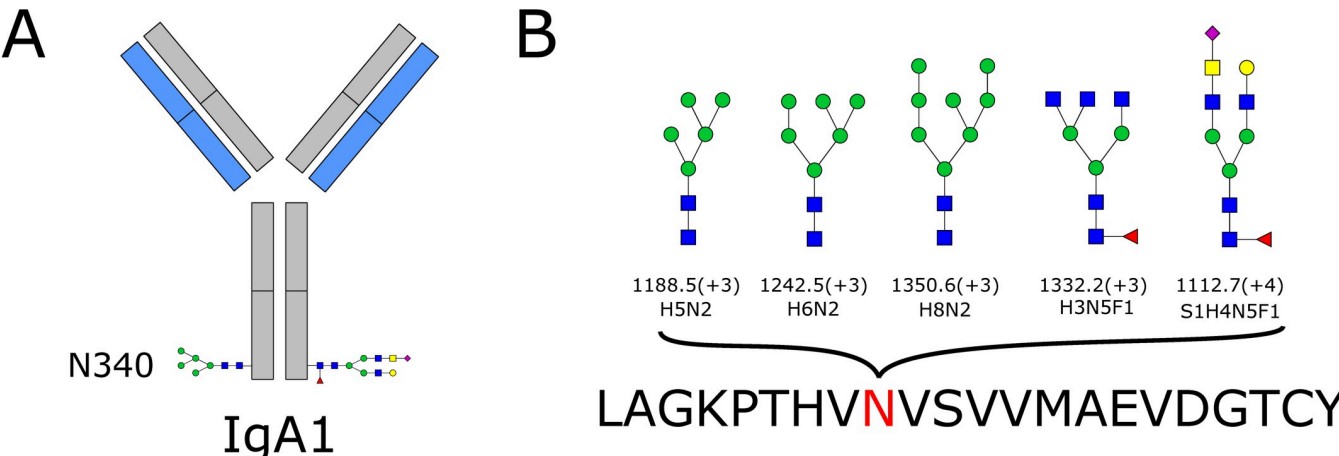

**Fig 1.** A) IgA1 with N-glycosylation site 340 shown. B) The five different glycan compositions identified at site 340 of IgA1. Five different N-glycopeptides were identified that belonged to IgA1 and contained glycans attached at site 340. Fig 1A shows IgA1 with two of these glycans as an example (it is not possible to determine if these two glycans were both detected on the same IgA1 antibody). The heavy chains are shown in gray and the light chains in blue. The three high-mannose N-glycans and two complex-type N-glycans identified at site 340 are shown in Fig 1B. Monosaccharide symbols follow the SNFG (Symbol Nomenclature for Glycans) system [18]. The m/z value and charge are given below the glycan structures. For the glycan structures, the following abbreviations were used: H = hexose, N = N-acetylhexosamine, F = fucose, S = sialic acid. The peptide sequence is given at the bottom of the figure, with asparagine (N) being the amino acid to which the glycan is linked. L = leucine, A = alanine, G = glycine, K = lysine, P = proline, T = threonine, H = histidine, V = valine, S = serine, M = methionine, E = glutamic acid, D = aspartic acid, C = cysteine, Y = tyrosine.

most glycan compositions identified was site 469, at which eight different glycans were identified. Different glycoforms of lactotransferrin (TRFL) were also identified, with a total of 24 different glycopeptides being identified. Of these, one was classified as a high-mannose glycan, one as a hybrid-type N-glycan, and two as monoantennary N-glycans. The remainder were complex-type N-glycans. Site 497 was the N-glycosylation site with the largest microheterogeneity, with a total of 11 different glycans identified at this site.

Of the nine glycan compositions identified that were part of glycopeptides originating from alpha-S1-casein, all but one contained fucose residues. Four of these glycans also contained sialic acid residues. In contrast, only two glycan compositions out of the eight glycopeptides identified that were part of IgA1 contained fucose residues, while one out of eight contained a sialic acid residue. Out of the 24 glycopeptides identified that were part of pIgR, more than half were fucosylated or sialylated, with 11 glycopeptides containing both a fucose residue and a sialic acid residue. The same was found true for lactotransferrin, with many of the glycopeptides identified being both fucosylated and sialylated. The glycan composition S1H5N4F2 was identified twice at site 69 on alpha-S1-casein. S1H5N4F2 was also identified twice at site 469 and twice at site 421 on pIgR. The glycan composition S1H5N4F1 was identified twice at site 421 on pIgR. In these cases, the same composition was found in different charged states and with different m/z values and subsequently identified twice at the same site.

## Differences between Rochester and OOM mothers

The samples analyzed in this study were first divided according to community, Rochester (n = 20) or OOM (n = 34). The 79 N-glycopeptides identified in this study were compared between these two cohorts and a total of 38 glycopeptides were found to have significantly different (p<0.05) abundances between the two groups. The top 10 glycopeptides according to fold change are given in Table 2 and all 38 glycopeptides are given in S2 Table. The majority of the glycan compositions identified were classified as complex-type N-glycans, although three glycans were classified as high-mannose N-glycans and two as monoantennary N-glycans. Out of the 38 N-glycopeptides, 26 were fucosylated and 22 were sialylated. Fifteen of 45 glycan compositions contained both fucose and sialic acid residues. The largest fold change (88.9) was observed for an N-glycopeptide originating from Immunoglobulin A2 (IGHA2, IgA2). The abundance of this N-glycopeptide, whose glycan composition was H3N5F1, was higher in the milk of Rochester mothers.

**Alpha-S1-casein.** The abundances of three of the nine N-glycopeptides identified at site 69 on alpha-S1-casein were significantly different when compared between Rochester and

**Table 2. The top 10 significant (p<0.05) glycopeptides according to fold change when samples were compared between Rochester mothers and OOM mothers.**
Nine out of 10 glycopeptides were also significant as analyzed by the Benjamini-Hochberg method. Further details can be found in S2 Table.

| Protein UniProt ID | N-glycosylation site | Peptide sequence | Glycan composition | Fold change | Mann-Whitney U test p-value |
|---|---|---|---|---|---|
| IGHA2_HUMAN | 205 | TPLTANITK | H3N5F1 | 88,93 | 3,45E-03 |
| FIBG_HUMAN | 78 | VDKDLQSLEDILHQVENK | S1H5N4F2 | 3,35 | 5,65E-04 |
| TRFL_HUMAN | 497 | TAGWNIPMGLLFNQTGSCK | S2H5N4F1 | 2,86 | 5,06E-04 |
| CASA1_HUMAN | 69 | NESTQNCVVAEPEK | S1H5N4F2 | 2,76 | 7,99E-03 |
| CASA1_HUMAN | 69 | NESTQNCVVAEPEK | S1H5N4F2 | 2,29 | 1,54E-02 |
| TRFL_HUMAN | 497 | TAGWNIPMGLLFNQTGSCK | S1H5N4F1 | 2,13 | 3,44E-04 |
| PIGR_HUMAN | 499 | WNNTGCQALPSQDEGPSK | S1H5N4 | 2,10 | 4,64E-06 |
| IGHA1_HUMAN | 340 | LAGKPTHVNVSVVMAEVDGTCY | H6N2 | 2,00 | 2,82E-03 |
| PIGR_HUMAN | 186 | QIGLYPVLVIDSSGYVNPNYTGR | S2H5N4 | 1,87 | 1,76E-04 |
| PIGR_HUMAN | 421 | LSLLEEPGNGTFTVILNQLTSR | S1H5N4F1 | 1,86 | 2,94E-02 |

OOM mothers. These three glycan compositions are given in S3 Table. Both glycan compositions were sialylated and classified as complex-type N-glycans, and the abundances of these glycopeptides were higher in samples from Rochester mothers.

**IgA.**    The abundances of five of the eight N-glycopeptides identified that belonged to IgA1 were significantly different when compared between Rochester and OOM mothers (S3 Table). Three were classified as complex-type N-glycans and two as high-mannose N-glycans. Four of the glycan compositions were located at site 340 of IgA1, with the remaining one located at site 144. The glycan composition located at site 144, H3N5, was classified as a complex-type N-glycan with a bisecting GlcNAc. One of the significantly different N-glycopeptides, S1H4N5F1, also contained a sialic acid residue. Further, we also identified an N-glycopeptide belonging to IgA2, which had a fold change of 88.9 between the groups and a higher abundance in Rochester mothers. The glycan composition of this glycopeptide was H3N5F1, which was classified as a complex-type N-glycan.

**pIgR.**    Out of the 24 N-glycopeptides identified that belonged to pIgR, given in Fig 2, the abundances of 14 glycopeptides, given in S3 Table, were significantly different between samples from Rochester and OOM mothers. At site 421, four glycan compositions were identified with different abundances between the groups. An N-glycopeptide containing the complex-type N-glycan S2H5N4F1 was identified at this site and had a fold change of 0.04 when compared between samples from Rochester and OOM mothers, indicating that levels of this N-glycopeptide were around 25 times higher in samples from OOM mothers. At site 469, four glycan compositions with significantly different abundances between the groups were identified, with three of these compositions displaying higher abundances in samples from OOM mothers. At site 499, five glycan compositions with different abundances between the groups were identified. Three of these five N-glycopeptides had significantly higher abundances in samples from Rochester mothers, among them an N-glycopeptide containing the high-mannose glycan H4N2. The glycopeptide containing the glycan composition S1H5N4 at site 499 was found to have a higher abundance in Rochester mothers, while the glycopeptide containing the composition S2H5N4 at site 499 had a higher abundance in OOM mothers.

**Lactotransferrin.**    Of the 24 different N-glycopeptides identified that belonged to lactotransferrin, the abundances of nine glycopeptides, given in S3 Table, were significantly different between samples from Rochester and OOM mothers. Eight of nine glycan compositions were located at site 497, while the ninth glycan was located at site 156. Interestingly, the four N-glycopeptides with higher abundances in Rochester mothers all contained sialylated glycans. The glycan compositions of the five N-glycopeptides with higher abundances in OOM mothers were all unsialylated. The N-glycopeptide containing the glycan composition H5N4F1 at site 497 had the largest fold change (8.9) between the groups.

## Differences between children with or without atopic disease

The samples in this study were also compared between mothers whose children developed atopic disease and those whose children did not (regardless of which community the mothers belonged to). Out of the 79 N-glycopeptides identified, the abundances of eight glycopeptides (given in Table 3) were significantly different among the two groups. Further details can be found in S4 Table. Five of nine glycan compositions were classified as complex-type N-glycans, while H8N2 was classified as a high-mannose N-glycan and H4N3F2 as a monoantennary N-glycan. The abundances of two glycopeptides were higher in samples from mothers whose children developed atopic disease, while the abundances of the remaining glycopeptides were higher in samples from mothers whose children did not develop atopic disease. The N-glycopeptide with the largest fold change (3.7) was identified on chordin-like protein 2 (CRDL2) and was more abundant in samples from mothers of children that developed atopic disease.

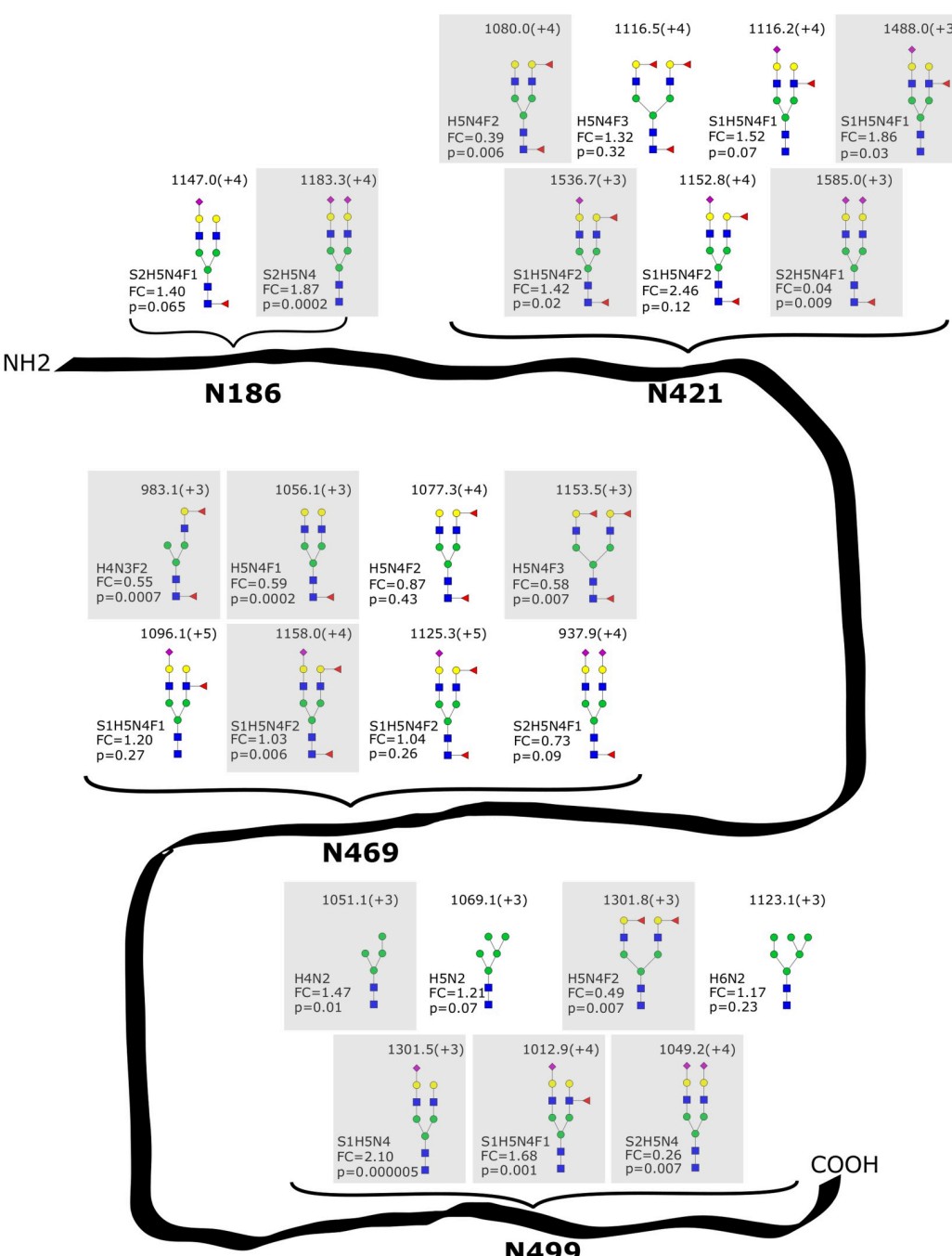

**Fig 2. The 24 glycan compositions identified belonging to N-glycopeptides that were part of pIgR.** Two glycan compositions were identified at site 186, seven at site 421, eight at site 469, and seven at site 499. The m/z value and charge are given above the glycan structures. Monosaccharide symbols follow the SNFG (Symbol Nomenclature for Glycans) system [18]. For the glycan structures, the following abbreviations were used: H = hexose, N = N-acetylhexosamine, F = fucose, S = sialic acid. Gray boxes indicate a significant (p<0.05) p-value. The fold change (FC) when abundances were compared between Rochester and OOM mothers is also given, with values greater than 1 indicating a higher abundance of the glycopeptide containing that specific glycan composition in Rochester mothers and values less than 1 indicating a higher abundance in OOM mothers. NH2 indicates the N-terminus and COOH the C-terminus of the protein.

**Table 3. The eight significant (p<0.05) glycopeptides identified, given according to fold change, when samples were compared between children who developed atopic disease and children who did not.** Further details can be found in S4 Table.

| Protein UniProt ID | N-glycosylation site | Peptide sequence | Glycan composition | Max fold change | Mann-Whitney U test p-value |
|---|---|---|---|---|---|
| CRDL2_HUMAN | 114 | SCQHNGTMYQHGEIFSAHELFPSR | S1H5N4F1 | 3,69 | 1,67E-02 |
| PIGR_HUMAN | 499 | WNNTGCQALPSQDEGPSK | S1H5N4 | 1,52 | 1,04E-02 |
| BT1A1_HUMAN | 55 | LSPNASAEHLELR | S2H5N4F1 | 0,76 | 4,30E-02 |
| PIGR_HUMAN | 469 | VPGNVTAVLGETLK | H4N3F2 | 0,75 | 3,86E-02 |
| IGHA1_HUMAN | 340 | LAGKPTHVNVSVVMAEVDGTCY | S1H4N5F1 | 0,45 | 2,20E-02 |
| HPT_HUMAN | 241 | VVLHPNYSQVDIGLIK | S2H5N4 | 0,42 | 1,11E-02 |
| PIGR_HUMAN | 499 | WNNTGCQALPSQDEGPSK | H5N4F2 | 0,37 | 3,95E-03 |
| IGHA1_HUMAN | 340 | LAGKPTHVNVSVVMAEVDGTCY | H8N2 | 0,32 | 2,33E-02 |

### Differences between atopic and non-atopic mothers

As significant differences were seen in the abundances of multiple N-glycopeptides when compared between samples from mothers whose children developed atopic disease (n = 9) versus those that did not (n = 42), we also decided to compare the samples between mothers with (n = 25) and without atopic disease (n = 26), regardless of community. The abundances of 18 out of 79 N-glycopeptides (given in S5 Table) were significantly different between samples from mothers with atopic disease and those without. Three glycan compositions were classified as high-mannose N-glycans and one as a monoantennary N-glycan, while the remainder were classified as complex-type N-glycans. The abundances of all but one N-glycopeptide were higher in mothers without atopic disease.

### Similarities between the comparisons

In Fig 3A, a Venn diagram comparing the significantly different N-glycopeptides in each comparison against each other is shown. Three N-glycopeptides were identified that had significantly different abundances in all three comparisons in this study (see Table 4 for details). The glycan compositions of these three N-glycopeptides are presented in Fig 3B. The abundances of these three N-glycopeptides were higher in samples from OOM mothers, samples from mothers whose children did not develop atopic disease, and samples from mothers without atopic disease. The N-glycopeptide containing the glycan composition H8N2, which was classified as a high-mannose N-glycan, was identified at site 340 on IgA1. The glycan composition H5N4F2, a complex-type N-glycan, was identified at site 499 on pIgR, while the composition H4N3F2, a monoantennary N-glycan, was identified at site 469 on pIgR. The proposed structures of these three glycan compositions matched entries in the GlyTouCan database.

The abundances of two N-glycopeptides were significantly different both when compared between samples from Rochester and OOM mothers and mothers whose children developed atopic disease versus mothers whose children did not, but not when compared between mothers with or without atopic disease. The N-glycopeptide with higher abundances in samples from Rochester mothers and samples from mothers whose children developed atopic disease was identified as belonging to pIgR. The glycan component of this glycopeptide, S1H5N4, was found at site 499 and classified as a complex-type N-glycan. The N-glycopeptide containing the glycan composition S1H4N5F1, also a complex-type glycan, had higher abundances in samples from OOM mothers and mothers whose children did not develop atopic disease. This glycan composition was found at site 340 of IgA1.

### Discussion

In this study, we analyzed a total of 54 milk samples that were divided according to community (Rochester or OOM) and, separately, the presence or absence of atopic disease in children and

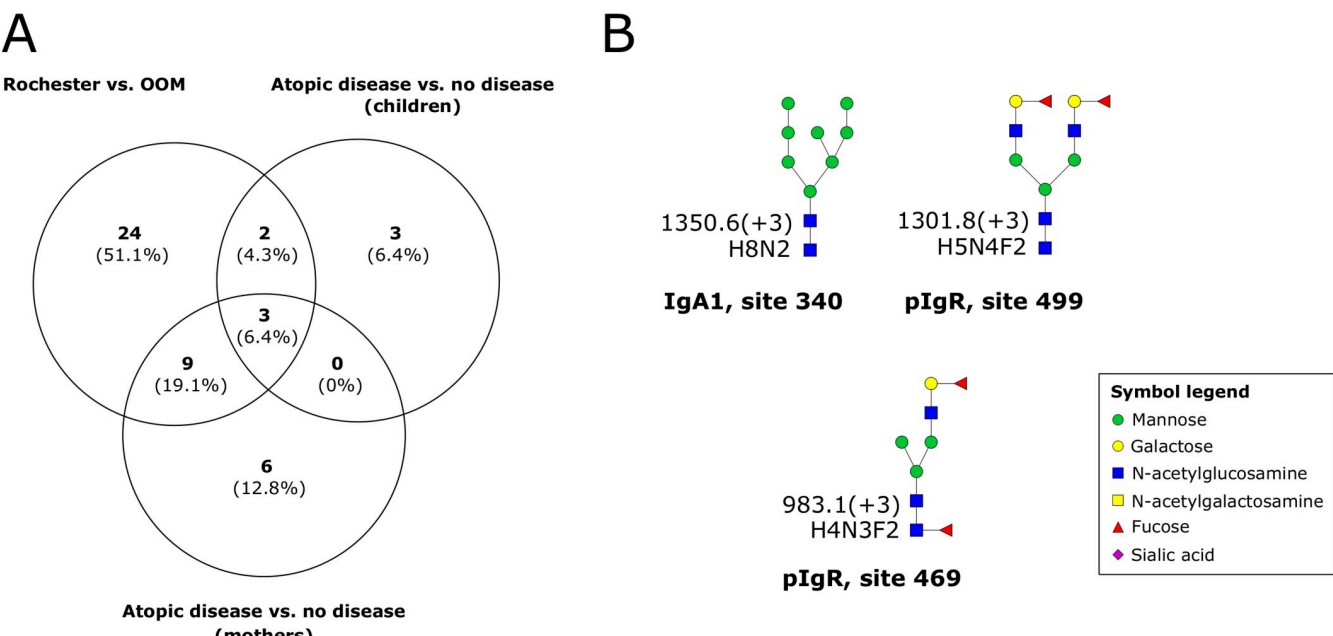

**Fig 3.** (A) The Venn diagram showing the significant N-glycopeptides identified in each comparison in relation to each other. (B) The three glycan compositions with significantly different abundances in all three comparisons in this study. Monosaccharide symbols follow the SNFG (Symbol Nomenclature for Glycans) system [18]. For the glycan structures, the following abbreviations were used: H = hexose, N = N-acetylhexosamine, F = fucose, S = sialic acid. The m/z value and charge are given to the left of the glycan structures. OOM = Old Order Mennonites. IgA1 = immunoglobulin A1. pIgR = polymeric immunoglobulin receptor.

mothers, regardless of community. The Rochester community represents a population at high risk of atopic disease, while the OOM community represents a population at low risk of atopic disease. A total of 38 N-glycopeptides had significantly different abundances when samples were compared according to community, while eight and 18 significantly different N-glyco-peptides were identified when samples were compared according to the presence or absence of atopic disease in children and mothers, respectively. These findings indicate that the significant differences seen in milk N-glycopeptide profiles between Rochester and OOM mothers, although to some degree be related to the development of atopic disease, are likely due to factors such as genetics, lifestyle, and exposures. The abundances of six of the nine N-glycopeptides identified when samples were compared according to the presence or absence of atopic disease in children were higher in children who did not develop atopic disease. This indicates that these glycopeptides are associated with protection against atopic disease in children.

**Table 4. The three N-glycopeptides identified that had significantly different abundances in all three comparisons (according to community, presence of atopic disease in children, and presence of atopic disease in mothers) in this study.**

| Protein UniProt ID | N-glycosylation site | Peptide sequence | Glycan composition | Rochester vs. OOM mothers | | Atopic disease vs. no disease (children) | | Atopic disease vs. no disease (mothers) | |
|---|---|---|---|---|---|---|---|---|---|
| | | | | Fold change | Mann-Whitney U test p-value | Fold change | Mann-Whitney U test p-value | Fold change | Mann-Whitney U test p-value |
| PIGR_HUMAN | 469 | VPGNVTAVLGETLK | H4N3F2 | 0,55 | 7,38E-04 | 0,75 | 3,86E-02 | 0,69 | 4,79E-03 |
| PIGR_HUMAN | 499 | WNNTGCQALPSQDEGPSK | H5N4F2 | 0,49 | 7,24E-03 | 0,37 | 3,95E-03 | 0,41 | 1,78E-04 |
| IGHA1_HUMAN | 340 | LAGKPTHVNVSVVMAEVDGTCY | H8N2 | 0,48 | 2,60E-02 | 0,32 | 2,33E-02 | 0,37 | 1,69E-03 |

Three N-glycopeptides were identified that had significantly different abundances in all three comparisons in this study. As the abundances of all three N-glycopeptides were higher in the OOM community and mothers and children without atopic disease, these findings indicate that these glycan compositions may reflect various environmental exposures but could be linked to the development of atopic disease and could affect the development of the infant's immune system. Two glycan compositions were identified on pIgR and the third on IgA. IgA is the most predominant immunoglobulin in human milk and is produced by B cells that have migrated to the mammary gland. pIgR is present at the basal surface of mammary epithelial cells, where it binds to IgA and is subsequently internalized into endosomes. This complex is then transported via transcytosis to the apical surface, where pIgR is proteolytically cleaved, leading to the release of secretory IgA (SIgA). SIgA, which consists of IgA and a large extracellular fragment of pIgR, the so-called secretory component, is then secreted into mammary alveoli [19, 20]. A study by Steffen et al. showed that altered glycosylation affects the effector functions of IgA. The authors found that the enzymatic removal of sialic acid increases the pro-inflammatory capacity of IgA1 [21]. Our findings indicate that the differential glycosylation of pIgR and IgA, two proteins that interact closely with each other, has a role in the development of atopic disease and may confer a protective effect against the development of atopic disease in both mothers and children. It is likely that the differences in glycosylation observed between IgA and pIgR in our study lead to functional differences, although the exact effects are unknown.

Strengths of this study include access to the OOM community, a unique population with an agrarian lifestyle and a low rate of allergic disease, and the large number of samples analyzed, which enabled the comparison of N-glycopeptide profiles between different groups and lends validity to the findings of our statistical analyses. The number of samples analyzed in our study is larger than what previous glycoproteomic studies of milk have used and, to the best of our knowledge, this is the first study to comprehensively analyze the N-glycoproteome between two communities at different risk of atopic disease. Further, in several cases, the same glycan composition, although with different charged states and m/z values, was identified twice at the same site of a protein. This serves to validate our findings, especially regarding the quantitation, which was independently done. As a limitation, the follow-up on development of allergic diseases was limited to self-reporting of physician diagnosis and/or a retrospective query of allergic symptoms performed by a pediatric allergist. The latter was implemented to mitigate the possible bias due to lower healthcare utilization by the OOM.

In conclusion, we show that the differential glycosylation of milk proteins may play a role in the development of atopic disease. We identified six N-glycopeptides that may protect against the development of atopic disease in children, and 16 that may protect against the development of atopic disease in mothers. Further, we identified three glycan compositions that may confer a protective effect against the development of atopic disease in both mothers and children.

## Materials and methods

### Study population

Milk samples from 54 mothers from a previously studied cohort [22] were analyzed. Of these, 20 samples came from Rochester mothers and 34 from OOM mothers. The OOM of the Finger Lakes region in New York State were recruited by a nurse midwife among prenatal visits in her clinic. Rochester urban and suburban mothers were recruited from the University of Rochester Medical Center using posted fliers. This study was approved by the Institutional Review Board of the University of Rochester Medical Center (RSRB52971) and all subjects provided written

consent prior to enrollment in the study. The samples used in this study are given in S6 Table together with relevant details.

Data were collected by questionnaires regarding maternal atopic disease (asthma, eczema, allergic rhinitis and food allergy), which was self-reported. In children, the presence of possible atopic disease in the first three years of life was determined through a blinded telephone follow-up for allergic symptoms, performed by a pediatric allergist (KJ). We queried physician-diagnosis of atopic dermatitis, allergic rhinitis, food allergy, and asthma, or symptoms consistent with atopic disease including chronic/remitting pruritic rash in a distribution age-typical for atopic eczema that had been treated by steroidal treatments, chronic or recurring symptoms of rhinorrhea/congestion/sneezing treated with antihistamines, symptoms suggestive of IgE-mediated food allergy (itching/swelling of lips/mouth/throat, urticaria or severe vomiting after 2h of ingestion of a specific food) and allergic proctocolitis, and recurrent wheezing episodes. Few intermittent wheezing episodes associated with viral infections alone were not determined indicative of asthma due to young age, whereas recurrent exercise-induced symptoms and persistent wheezing treated with inhaled corticosteroids were labeled as "recurrent wheezing/asthma". Some data regarding the presence or absence of atopic disease in mothers and children was unavailable (marked N/A in S6 Table) due to a lack of follow-up data.

## Sample collection

Milk samples were collected in the morning by the use of UV-irradiated sterile manual breast pumps (Harmony Breastpump, Medela) or manual expression, wearing gloves. Foremilk was collected after cleaning the breast with Castille soap. All milk samples were frozen at -20°C immediately upon collection and transferred to -80°C within four weeks for storage.

## Trypsin digestion

Milk samples were defatted by centrifuging at 2,000g for 20 minutes at 4°C, after which the supernatant was transferred to a new tube and centrifuged at 5,000g for 10 minutes at 4°C. The supernatant was collected and transferred to a new tube again. This supernatant was then used for BCA assay. Equivalent amounts of protein were adjusted to the same concentration and diluted 1:1 with PBS (pH 7.4) and 6 volumes of cold acetone was added to the samples. After vortexing to mix the samples, the tubes were incubated at -20°C for 2 hours. The samples were centrifuged at 10,000g for 30 minutes at 4°C and the supernatant was carefully decanted. Pellets were dissolved in 50 mM Tris buffer + 8M urea (pH 8.0) and 10mM dithiothreitol (final concentration) was added to the samples. Samples were reduced for 1 hour at RT with mixing. Iodoacetamide was added to the samples at 40mM (final concentration) and the samples were alkylated at RT for 1 hour with shaking in the dark. Further, 40mM DTT was added to prevent overalkylation by iodoacetamide. Bovine pancreatic trypsin was added to the solution at a 1:50 ratio and the samples were incubated at 37°C overnight.

## Lectin affinity chromatography

Lectin affinity chromatography was performed as described previously [23]. Briefly, tryptic peptides were diluted 1:10 with 10mM HEPES buffer (pH 7.4) containing 1mM CaCl2 and 1mM MnCl2. Diluted sample mixtures were individually incubated with lectin-agarose columns. Con-A:SNA:LCA:AAL (5:3:3:1) were used for a final volume of 150 μL lectin resin slurry, which was mixed with samples and incubated overnight at 4°C with rotation. 24 hours later, lectin beads were washed three times with HEPES buffer and the N-glycopeptides were eluted with a solution containing fucose, α-methyl mannoside, α-methyl glucoside, and lactose, followed by 1% formic acid. The N-glycopeptides were cleaned using C18 micro spin

columns according to the manufacturer's instructions. 0.1% formic acid was used to dissolve the eluted N-glycopeptides for analysis by Ultra-Performance Liquid Chromatography (UPLC)-mass spectrometry (MS).

## UPLC-MS/MS2 and -MSE

N-glycopeptides were analyzed using a Waters SYNAPT G2 High Definition MS connected to a Waters nanoACQUITY UPLC. MSE (100–2000 Da mass range) was performed in positive mode with sensitivity mode for the quantification and MS2 FAST DDA (positive and sensitivity) mode was used for N-glycopeptide fragmentation (50–2500 Da mass range) for identification. One second scan time was used for both MSE and FAST DDA. Trap collision energy (high energy function) was ramped from 14 to 44 V for MSE. Continuum data format and deisotope peak selection was set in the parameters. In the FAST DDA trap collision, energy was ramped with low mass to high mass range from 20 to 60 V. Calibration was performed with sodium formate. The trap column was a nanoACQUITY UPLC Trap, 180 μm x 20 mm (5 μm), Symmetry®C18, and the analytical column was a nanoACQUITY UPLC, 75 μm x 100 mm (1.8 μm), HSS T3. Samples were loaded, trapped, and washed for two minutes with 8.0 μL/min with 1% B. The analytical gradient used is as follows: 0–1 minutes 1% B, at 2 minutes 5% B, at 45 minutes 30% B, at 48 minutes 50% B, at 50 minutes 85% B, at 53 minutes 85% B, at 54 minutes 1% B and at 60 minutes 1% B with 450nL/min for MSE while 300 nL/min was used for N-glycopeptide fragmentation.

## Data analysis

The raw files were imported to Progenesis QI for proteomics (Version V2, Nonlinear Dynamics, Newcastle, UK). Post-acquisition mass correction was performed when the files were imported using a lock mass correction of 785.8426 m/z (doubly-charged Glu1-Fibrinopeptide B). The default parameters for peak picking and alignment were used. Progenesis QI for proteomics performs the label-free quantification and works as follows:

**Run alignment.** The run with the most peaks (ions) is used as a reference to which all other runs are aligned.

**Peak picking.** All data from the aligned runs is aggregated into a dataset that contains all peak information from all sample files. This aggregate peak list is then used to match each individual sample file.

**Ion abundance quantification.** Peptide ion abundance is a sum of areas that are obtained from calculating the peak areas from the intensity curves obtained from the MSE runs. Charge selection is applied by considering +3 to +5 charged ions as potential N-glycopeptides. The groups are then compared to show differences between the groups and finally, MS/MS is matched to MSE runs to collate quantitation and identification information.

N-glycopeptide ions were identified as previously described [23]. Briefly, deconvolution of the MS/MS spectra was performed using the MaxEnt3 module of Waters MassLynx 4.1 software and exported as peak lists (.pkl). The publicly available software GlycopeptideID was used to identify the N-glycopeptides, as GlycopeptideID can analyze CID MS/MS spectra in an automated manner. A database of tryptic peptides (Uniprot, two miscleavages allowed, mandatory to have NXS/T/C, P! in the peptide sequence, where X is any amino acid other than proline) from known human proteins was used and a glycan library was generated within the software by downloading human glycans from the GlyTouCan glycan structure repository (https://glytoucan.org). All deconvoluted MS/MS spectra are imported as.pkl files combined into one file. MS2 spectra are matched against this database and scores for potential peptide backbones are generated. GlycopeptideID then searches glycan compositions against a glycan

database and returns glycan compositions which are fitted onto the spectrum by generating glycan scores. Peptide and glycan scores are summed up to provide glycopeptide scores [24]. The results are then ranked and an annotated spectrum is drawn for each potential result. Manual assessment of the matching y and b ions, glycan fragments, and glycopeptide fragments is subsequently performed. The target-decoy strategy is used to calculate the false-discovery rate. The decoy database was generated by reversing the peptide sequences of the tryptic peptide database specified above. The false-discovery rate was set to 2%.

## Glycan compositions

Glycan compositions are specified as one-letter abbreviations: H = hexose, N = hexosamine, S = sialic acid, and F = fucose. The number following the abbreviations indicates the number of monosaccharides. For example, S2H6N5 indicates a glycan containing two sialic acids, six hexoses, and five hexosamines. A simplified version of the consortium for functional genomics (CFG) Modified IUPAC condensed format is used for glycan structure output (see www.functionalglycomics.org/static/consortium/Nomenclature.shtml for details. The stereoisomer (α, β) and the region isomer (for ex. 1–4) notations are removed and single-letter codes (H = Hex, N = HexNAc, F = Fuc, S = NeuAc) are used for monosaccharides. Branching is shown by parenthesis and written from non-reducing to reducing end. As the glycan compositions identified matched database entries in the GlyTouCan repository, it was possible to provide proposed glycan structures for these compositions. Glycans were also divided into complex-type, high-mannose, hybrid, or monoantennary N-glycans on the basis of their proposed structures.

## Statistical analysis

The samples were divided into groups based on community (Rochester or OOM), as well as the presence or absence of atopic disease in the children and mothers (see S6 Table). The Mann-Whitney U test was used to analyze the differences between the groups and a p-value of <0.05 was considered significant.

## Supporting information

**S1 Table. The 79 N-glycopeptides identified in this study.**
(XLSX)

**S2 Table. The 38 N-glycopeptides with significantly different abundances when samples were compared according to community (Rochester vs. OOM).**
(XLSX)

**S3 Table. The N-glycopeptides identified from alpha-S1-casein, IgA, pIgR, and lactotransferrin with significantly different abundances when samples were compared according to community (Rochester vs. OOM).**
(XLSX)

**S4 Table. The eight N-glycopeptides with significantly different abundances when samples were compared between children who developed atopic disease and those that did not.**
(XLSX)

**S5 Table. The 18 N-glycopeptides with significantly different abundances when samples were compared between mothers with or without atopic disease.**
(XLSX)

**S6 Table. The milk samples analyzed in this study together with details regarding the mother's community and the presence or absence of atopic disease in both mothers and children.**
(XLSX)

## Acknowledgments

The authors would like to thank Mary Ann Martin and the late Joyce Wade, CNM for their guidance and assistance in recruitment efforts within the Old Order Mennonite Community and the participating families without whom this research would not be possible.

## Author Contributions

**Conceptualization:** Risto Renkonen, Kirsi M. Järvinen.

**Data curation:** Mayank Saraswat, Sakari Joenväärä, Antti Seppo.

**Formal analysis:** Mayank Saraswat, Sakari Joenväärä.

**Funding acquisition:** R. John Looney, Risto Renkonen, Kirsi M. Järvinen.

**Investigation:** Mayank Saraswat, Sakari Joenväärä, Tiialotta Tohmola, Jutta Renkonen, Kirsi M. Järvinen.

**Methodology:** Antti Seppo, R. John Looney, Kirsi M. Järvinen.

**Project administration:** Risto Renkonen, Kirsi M. Järvinen.

**Resources:** Antti Seppo, Risto Renkonen, Kirsi M. Järvinen.

**Software:** Sakari Joenväärä.

**Supervision:** Risto Renkonen, Kirsi M. Järvinen.

**Validation:** Sakari Joenväärä.

**Visualization:** Matilda Holm, Sakari Joenväärä.

**Writing – original draft:** Matilda Holm, Kirsi M. Järvinen.

**Writing – review & editing:** Mayank Saraswat, Sakari Joenväärä, Antti Seppo, R. John Looney, Tiialotta Tohmola, Jutta Renkonen, Risto Renkonen, Kirsi M. Järvinen.

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
