## [Decision Letter · Decision Letter 0]

8 Nov 2021

PONE-D-21-30987Quantitative glycoproteomics of human milk and association with atopic diseasePLOS ONE

Dear Dr. Jarvinen,

Thank you for submitting your manuscript to PLOS ONE. After careful consideration, we feel that it has merit but does not fully meet PLOS ONE’s publication criteria as it currently stands. Therefore, we invite you to submit a revised version of the manuscript that addresses the points raised during the review process.

Both reviewers expressed concerns in finding clear explanations on some key points such as quantification and the 2nd review stresses more concerns on precision and rigor lacking in data analysis and interpretation that are crucial steps in the study.

We look forward to receiving your revised manuscript.

Kind regards,

Frederique Lisacek

Academic Editor

PLOS ONE

Journal Requirements:

Reviewers' comments:

Reviewer's Responses to Questions

**Comments to the Author**

1. Is the manuscript technically sound, and do the data support the conclusions?

Reviewer #1: Yes

Reviewer #2: No

2. Has the statistical analysis been performed appropriately and rigorously? 

Reviewer #1: Yes

Reviewer #2: I Don't Know

3. Have the authors made all data underlying the findings in their manuscript fully available?

Reviewer #1: Yes

Reviewer #2: Yes

4. Is the manuscript presented in an intelligible fashion and written in standard English?

Reviewer #1: Yes

Reviewer #2: Yes

5. Review Comments to the Author

Reviewer #1: Overall, this is a well performed study and is well described. I think that it provides a nice set of information for the field which may lead to future research examining the potential impact of the associations between site-specific glycosylation of milk proteins and allergy.

Abstract

32: N-glycopeptide should be N-glycoprotein—as you are using tryptic peptides to identify the glycoproteins.

34: same comment. Should be glycoprotein, not glycopeptide profiles. Saying glycopeptide implies that you studied the naturally occurring glycopeptides rather than the trypsin-produced peptides.

Intro:

93: you are actually comparing N-glycoprotein profiles (via tryptic N-glycopeptides), not N-glycopeptide profiles—the N-glycopeptides here are created in your study, not a natural phenomenon.

Results

Table 2: max fold change and P-value numbers shoul have decimal points as periods, not commas. Same for Table 3.

Method

424: Should give time and temperatures for centrifugation steps.

433: iodoacetamide should not be capitalized.

Statistics: It seems that a correction for multiple testing should be applied (each test increases the likelihood of a false positive). I do not see one listed.

Reviewer #2: Data interpretation: The abstract is overstretched. “These four glycopeptides may have a protective effect against the development of atopic disease. Our findings indicate that the differential glycosylation of milk proteins may affect the development of atopic disease, something previously uninvestigated.” As the authors do present a mere association, yet no mechanistic hypothesis, and also no experimental support for a causal relationship whatsoever, these final sentences of the abstract appear poorly supported. Please adjust.

Data interpretation: what are the differences between the milk samples? Please try an distill from the data some major changes: how is site-specific glycosylation different, is e.g. sialylation up or down, are there differences in fucosylation?

To which extent are the differences in glycopeptide signals caused by differences in the expression of proteins, versus differences in site-specific glycosylation? An integrated proteomic and glycoproteomic analysis of the data would be necessary.

Glycopeptide identification: The authors mention that MS/MS assignments were manually checked, but they do not display any of the MS/MS spectra. Please provide assigned MS/MS spectra as supplementary data for all de novo assignments and for all more exotic compositions and proposed structures.

Figure 1: the H3N3S1 glycan structure is not credible, with the sialic acid linked to GlcNAc; please adjust

Figure 2: what is the structural basis for assigning fucoses to the core, to the antennary GlcNAc, or antennary galactose; please provide support for these assignments, same for figure 3.

Quantification: quantifying each charge state separately does not make sense, as the aim is to quantify glycopeptides, not MS signals. Hence, please sum up the different charge states before quantification, or rely on the most intense one.

The quantification needs some clarification: For example, it is unclear how the fold change was calculated, as the tables have a column headed Max fold change. What is a Max fold change ? I searched the manuscript for explanation, but could not find it.

In line with this comment, please provide comprehensive legends for all suppl. Tables to guide the reader in assessing the data.

6. PLOS authors have the option to publish the peer review history of their article (what does this mean?). If published, this will include your full peer review and any attached files.

Reviewer #1: No

Reviewer #2: No

---

## [Decision Letter · Decision Letter 1]

28 Feb 2022

PONE-D-21-30987R1Quantitative glycoproteomics of human milk and association with atopic diseasePLOS ONE

Dear Dr.Jarvinen,

Thank you for submitting your manuscript to PLOS ONE. After careful consideration, we feel that it has merit but does not fully meet PLOS ONE’s publication criteria as it currently stands. Therefore, we invite you to submit a revised version of the manuscript that addresses the points raised during the review process. The revised manuscript still does not include enough evidence supporting some claims in the text. This requires according to reviewer #2, the enhancement of supplementary material and the inclusion of further details justifying some structural assignments.

We look forward to receiving your revised manuscript.

Kind regards,

Frederique Lisacek

Academic Editor

PLOS ONE

Journal Requirements:

Reviewers' comments:

Reviewer's Responses to Questions

**Comments to the Author**

1. If the authors have adequately addressed your comments raised in a previous round of review and you feel that this manuscript is now acceptable for publication, you may indicate that here to bypass the “Comments to the Author” section, enter your conflict of interest statement in the “Confidential to Editor” section, and submit your "Accept" recommendation.

Reviewer #1: All comments have been addressed

Reviewer #2: (No Response)

2. Is the manuscript technically sound, and do the data support the conclusions?

Reviewer #1: Yes

Reviewer #2: (No Response)

3. Has the statistical analysis been performed appropriately and rigorously? 

Reviewer #1: Yes

Reviewer #2: I Don't Know

4. Have the authors made all data underlying the findings in their manuscript fully available?

Reviewer #1: Yes

Reviewer #2: No

5. Is the manuscript presented in an intelligible fashion and written in standard English?

Reviewer #1: Yes

Reviewer #2: Yes

6. Review Comments to the Author

Reviewer #1: All changes requested have been made accurately. The manuscript is now ready for publication in the journal.

Reviewer #2: The authors have considerably improved the manuscript by toning down on some statements, and including more information in e.g figure and table legends. A lot of the key data of the paper are, however, not sufficiently accessible, and I therefore propose a major revision of the manuscript, to make some of the conclusions more substantiated, make the paper more comprehensible, and ultimately increase its impact.

1. Display of MS/MS data. The authors indicate that MS/MS data are made online available. I think this is not sufficient. For the most "exotic" assignments the authors should provide fully assigned spectra to support their conclusions. I would e.g. love to see the assigned MS/MS spectrum of S2H3N10F5 now indicated in Figure 2. Also for the small number of additional, de novo assigned glycopeptides, a display of the assigned spectra may help to support the claims.

2. Another aspect of concern is the use of "Max fold change". I think one can safely agree that "Max fold change" is not a conventional way of assessing differences between groups, and is not the most meaningful parameter. Obviously, I am not interested in seeing the maximum difference between groups, but the differences of medians or means in these cross-sectional comparisons. Displaying such averages and ranges (e.g. interquartile ranges, or standard deviations) would make the paper biologically way more meaningful.

3. As a reader I would need more relevant information in the supplemental tables: what is the score column? Were the p-values corrected? If so, in which manner, with which factor? Which significance threshold was applied?

4. For some glycopeptides the antennary fucoses were assigned to galactoses, for other to N-acetylglucosamine. I would love to see the data that support this assignment, in the form of some exemplary MS/MS spectra. I would be particularly interested to know whether the authors saw diagnostic B ions such as fucose-hexose, that would substantiate such a claim.

7. PLOS authors have the option to publish the peer review history of their article (what does this mean?). If published, this will include your full peer review and any attached files.

Reviewer #1: No

Reviewer #2: No

---

## [Author Response · Author response to Decision Letter 1]

30 Mar 2022

Replies to reviewer comments

We thank the reviewers for their suggestions. We have addressed all the comments as followes.

Reviewer #1: All changes requested have been made accurately. The manuscript is now ready for publication in the journal.

Reviewer #2: The authors have considerably improved the manuscript by toning down on some statements, and including more information in e.g figure and table legends. A lot of the key data of the paper are, however, not sufficiently accessible, and I therefore propose a major revision of the manuscript, to make some of the conclusions more substantiated, make the paper more comprehensible, and ultimately increase its impact.

1. Display of MS/MS data. The authors indicate that MS/MS data are made online available. I think this is not sufficient. For the most "exotic" assignments the authors should provide fully assigned spectra to support their conclusions. I would e.g. love to see the assigned MS/MS spectrum of S2H3N10F5 now indicated in Figure 2. Also for the small number of additional, de novo assigned glycopeptides, a display of the assigned spectra may help to support the claims.

We have now provided the PRIDE database ID for each glycan composition in the respective tables to make it easier to access the mass spectrometric data. The spectra and fragments were also previously accessible in the database. For clarity and simplicity, we have removed the de novo compositions from this study. 

2. Another aspect of concern is the use of "Max fold change". I think one can safely agree that "Max fold change" is not a conventional way of assessing differences between groups, and is not the most meaningful parameter. Obviously, I am not interested in seeing the maximum difference between groups, but the differences of medians or means in these cross-sectional comparisons. Displaying such averages and ranges (e.g. interquartile ranges, or standard deviations) would make the paper biologically way more meaningful.

We have now removed the maximum fold changes and replaced them with fold changes calculated using the difference of the mean abundance of each glycopeptide between the groups compared. 

3. As a reader I would need more relevant information in the supplemental tables: what is the score column? Were the p-values corrected? If so, in which manner, with which factor? Which significance threshold was applied?

The glycopeptide scores in the supplemental tables are calculated as a negative logarithm of the probability that a random set of fragments would have as many or more shared peaks with the measured spectrum as the ranked glycopeptide. The probability that the random spectra have more or equal shared peaks than the glycopeptide spectrum is calculated using binomial distribution. The glycopeptide scores therefore provide an indication of the quality of the identification. We have now referred to the study explaining this in detail in the Material and methods section. 

The p-values provided in the tables in the manuscript are uncorrected, with p-values of <0.05 being considered significant. We also used the Benjamini-Hochberg procedure to control the false-discovery rate and provide information regarding if the adjusted p-values were significant or not in the supplementary tables. 

4. For some glycopeptides the antennary fucoses were assigned to galactoses, for other to N-acetylglucosamine. I would love to see the data that support this assignment, in the form of some exemplary MS/MS spectra. I would be particularly interested to know whether the authors saw diagnostic B ions such as fucose-hexose, that would substantiate such a claim.

The MS/MS spectra of all fucosylated glycopeptides were manually examined in order to see if, in the case of fucose residues assigned to the core, we could identify primarily and secondarily peptide-HexNAc-fucose ions so that monosaccharides instead of HexNAc residues could be assigned to the N-core, as only one fucose can be assigned to the N-core. In the case of fucose residues assigned to branches, we looked for fucose-hexose-HexNAc (FHN) ions. 

In cases where fucose residues were only assigned to the core, no FHN ions were present in any of the spectra. In cases where fucose residues were assigned to branches, the software identified diagnostic FHN ions in 28 cases but missed 3 cases. In all cases, the final glycan composition used is the one with the best fit as calculated by fitting the total mass of the glycan composition, all the available intact peptide-glycan fragments, glycan fragments, and peptide-partial glycan fragments to the peptide whose sequence was assigned to that glycan. However, we would like to mention that wherever mentioned, we have only discussed proposed glycan structures.

---

## [Decision Letter · Decision Letter 2]

20 Apr 2022

Quantitative glycoproteomics of human milk and association with atopic disease

PONE-D-21-30987R2

Dear Dr. Jarvinen,

We’re pleased to inform you that your manuscript has been judged scientifically suitable for publication and will be formally accepted for publication once it meets all outstanding technical requirements.

Kind regards,

Frederique Lisacek

Academic Editor

PLOS ONE

Additional Editor Comments (optional):

Reviewers' comments:

Reviewer's Responses to Questions

**Comments to the Author**

1. If the authors have adequately addressed your comments raised in a previous round of review and you feel that this manuscript is now acceptable for publication, you may indicate that here to bypass the “Comments to the Author” section, enter your conflict of interest statement in the “Confidential to Editor” section, and submit your "Accept" recommendation.

Reviewer #2: All comments have been addressed

2. Is the manuscript technically sound, and do the data support the conclusions?

Reviewer #2: Partly

3. Has the statistical analysis been performed appropriately and rigorously? 

Reviewer #2: Yes

4. Have the authors made all data underlying the findings in their manuscript fully available?

Reviewer #2: Yes

5. Is the manuscript presented in an intelligible fashion and written in standard English?

Reviewer #2: Yes

6. Review Comments to the Author

Reviewer #2: (No Response)

7. PLOS authors have the option to publish the peer review history of their article (what does this mean?). If published, this will include your full peer review and any attached files.

Reviewer #2: No

---

## [Editor Report · Acceptance letter]

27 Apr 2022

PONE-D-21-30987R2 

Quantitative glycoproteomics of human milk and association with atopic disease 

Dear Dr. Jarvinen:

I'm pleased to inform you that your manuscript has been deemed suitable for publication in PLOS ONE. Congratulations! Your manuscript is now with our production department. 

Kind regards, 

on behalf of

Dr. Frederique Lisacek 

Academic Editor

PLOS ONE